# Maternal and Infant Histo-Blood Group Antigen (HBGA) Profiles and Their Influence on Oral Rotavirus Vaccine (Rotarix^TM^) Immunogenicity among Infants in Zambia

**DOI:** 10.3390/vaccines11081303

**Published:** 2023-07-31

**Authors:** Adriace Chauwa, Samuel Bosomprah, Natasha Makabilo Laban, Bernard Phiri, Mwelwa Chibuye, Obvious Nchimunya Chilyabanyama, Sody Munsaka, Michelo Simuyandi, Innocent Mwape, Cynthia Mubanga, Masuzyo Chirwa Chobe, Caroline Chisenga, Roma Chilengi

**Affiliations:** 1Enteric Disease and Vaccine Research Unit, Centre for Infectious Disease Research in Zambia, Lusaka P.O. Box 34681, Zambia; samuel.bosomprah@cidrz.org (S.B.); natasha.laban@cidrz.org (N.M.L.); bernard.phiri@cidrz.org (B.P.); mwelwa.chibuye@cidrz.org (M.C.); chilyabanyama@gmail.com (O.N.C.); michelo.simuyandi@cidrz.org (M.S.); innocent.mwape@cidrz.org (I.M.); cynthia.mubanga@cidrz.org (C.M.); masuzyo.chirwa@cidrz.org (M.C.C.); caroline.chisenga@cidrz.org (C.C.); roma.chilengi@cidrz.org (R.C.); 2Department of Biomedical Sciences, School of Health Sciences, University of Zambia, Lusaka P.O. Box 50110, Zambia; s.munsaka@unza.zm; 3Department of Biostatistics, School of Public Health, University of Ghana, Accra P.O. Box LG13, Ghana; 4Department of Infection Biology, Faculty of Infectious and Tropical Diseases, London School of Hygiene and Tropical Medicine, London WC1E 7HT, UK; 5Department of Global Health, Amsterdam Institute for Global Health and Development (AIGHD), Amsterdam University Medical Centers, University of Amsterdam, 1012 WP Amsterdam, The Netherlands

**Keywords:** rotavirus, vaccines, histo-blood groups, immunogenicity, Zambia

## Abstract

Live-attenuated, oral rotavirus vaccines have significantly reduced rotavirus-associated diarrhoea morbidity and infant mortality. However, vaccine immunogenicity is diminished in low-income countries. We investigated whether maternal and infant intrinsic susceptibility to rotavirus infection via histo-blood group antigen (HBGA) profiles influenced rotavirus (ROTARIX^®^) vaccine-induced responses in Zambia. We studied 135 mother–infant pairs under a rotavirus vaccine clinical trial, with infants aged 6 to 12 weeks at pre-vaccination up to 12 months old. We determined maternal and infant ABO/H, Lewis, and secretor HBGA phenotypes, and infant FUT2 HBGA genotypes. Vaccine immunogenicity was measured as anti-rotavirus IgA antibody titres. Overall, 34 (31.3%) children were seroconverted at 14 weeks, and no statistically significant difference in seroconversion was observed across the various HBGA profiles in early infant life. We also observed a statistically significant difference in rotavirus-IgA titres across infant HBGA profiles at 12 months, though no statistically significant difference was observed between the study arms. There was no association between maternal HBGA profiles and infant vaccine immunogenicity. Overall, infant HBGAs were associated with RV vaccine immunogenicity at 12 months as opposed to in early infant life. Further investigation into the low efficacy of ROTARIX^®^ and appropriate intervention is key to unlocking the full vaccine benefits for U5 children.

## 1. Introduction

Rotavirus is known to be the leading cause of moderate-to-severe acute gastroenteritis in infants and children under the age of 5 years (U5) globally, but more so in low- and middle-income countries (LMICs) [1]. In 2017, the Global Burden of Disease (GBD) study estimated total diarrhoea deaths in the U5 population attributable to RV to be between 120,000 and 215,000 [2]. Vaccines against rotavirus, such as ROTARIX^®^ (GlaxoSmithKline Biologicals, Rixensart, Germany), a G1(P8) strain-derived live-attenuated oral vaccine, have been rolled out in a national expanded program on immunisation (EPI) schedules in many LMICs, including Zambia in 2013, as recommended by the World Health Organisation (WHO) [3]. Despite successes in reducing rotavirus-associated and all-cause acute gastroenteritis recorded over the years [4], oral vaccine immunogenicity is diminished in LMIC settings where the burden of disease and need for such interventions is greatest in contrast to high-income countries [5,6].

Host genetic factors may influence rotavirus vaccine immunogenicity [7,8,9,10]. Recent studies have shown the role of histo-blood group antigens (HBGAs) as cell receptors utilised by rotavirus during infection of the host’s mucosal epithelium [11]. These HBGAs have been shown to mediate rotavirus infection in a P-genotype-specific manner, and this has been shown to have the potential to influence vaccine uptake, and, consequently, the efficacy of vaccines based on the G1(P8) live-attenuated strain [12,13,14,15,16,17]. Investigating HBGAs and their potential influence on vaccine immunogenicity provides actionable information that would accelerate efforts to improve vaccine efficacy in U5 children in LMICs.

Histo-blood group antigens include the blood group ABH and Lewis antigen systems, which are encoded by fucosyltransferase-2 (FUT2) and fucosyltransferase-3 (FUT-3) genes, respectively. In addition to red blood cells, these antigens can also be present in other body fluids such as saliva, breast milk, urine, seminal fluid, and other gastric secretions [18,19]. Currently, only a few studies have been conducted in African settings to investigate the role of HBGAs in susceptibility to rotavirus-induced AGE and rotavirus vaccine immunogenicity in children [12,14,16,20], and fewer still have been conducted on HBGAs in breastfeeding mothers [21,22,23]. We aimed to profile maternal and infant HBGA phenotypes and genotypes and determine their influence on ROTARIX^®^ immunogenicity in a mother–infant pair cohort in Zambia.

## 2. Materials and Methods

### 2.1. Study Design and Participants

This was a prospective cohort study of mother–infant pairs nested under a parent-randomised controlled trial (RCT). The parent study aimed to determine the safety and immunogenicity of a third booster dose of ROTARIX^®^ at 9 months of age, as published elsewhere [24]. Briefly, this study was conducted at a government health facility serving a peri-urban population in Lusaka, Zambia. The parent study enrolled 214 infants aged between 6 and 12 weeks with informed consent obtained from willing mothers who met the full eligibility criteria and agreed to all study procedures throughout the study. In addition to receiving the routinely administered first and second doses of ROTARIX^®^, infants were randomised at baseline at a ratio of 1:1 to either the intervention arm receiving a booster dose of ROTARIX^®^ concomitantly with measles/rubella (MR) vaccination, or the control arm receiving only MR vaccination, at 9 months old.

For this study, we randomly selected 135 participants from the parent study using a simple random sampling technique in Stata 17 (StataCorp, College Station, TX, USA). We determined the HBGA phenotype profiles of these selected participants and evaluated ROTARIX^®^ immunogenicity by analysing rotavirus-specific IgA antibody responses at various time points. Further, a random sample of 90 was selected for FUT2 blood buffy-coat genotyping from the 135 samples using the same method as that used for phenotype selection.

### 2.2. Laboratory Testing

#### 2.2.1. Determination of the Infant ABO and Lewis HBGA Phenotypes in Saliva

The blood groups A, B, O, H, Lewis a and b HBGA, and Lectin (Ulex europaeus agglutinin-1) were detected in saliva using an enzyme-linked immunosorbent assay (ELISA) adapted from previously described methods [16,25]. Briefly, samples diluted in buffer were incubated at 37 °C, followed by incubation at 4 °C overnight. The following day, the plate was blocked with 5% Blotto in TBS (Cat#: 786-192, BLOCK^TM^, G-BIOSCIENCES^®^, St. Louis, MO, USA) and incubated at 37 °C for 1 h, and later incubated with appropriate antibodies at 37 °C (Anti-Lewis a antibody [7LE] (Cat#: ab3967, Abcam, Cambridge, UK), Anti-Lewis b antibody [2-25LE] (Cat#: 922302, Abcam, Cambridge, UK), Anti-Blood Group A Antigen antibody [9A] (Cat#: ab20131, Abcam, Cambridge, UK), Mouse Anti-Blood Group B Antigen antibody (Cat#: ab24224, Abcam, Cambridge, UK), and Ulex europaeus1 Lectin (Cat#: L8146-1MG, Sigma-Aldrich, St. Louis, MO, USA). Next, the plate was incubated with Goat Anti-Mouse Immunoglobulin G (IgG) H&L Horse-radish peroxidase (HRP) conjugate (Cat#: ab48386, Abcam), and the reaction was developed using a chromogenic substrate in a dark cupboard at room temperature for 15 min. The reaction was stopped using sulphuric acid, while absorbance was read at 450 nm on an ELISA plate reader. Similarly, the ELISA method described for saliva above was used to detect Lewis and secretor phenotypes in breast milk with the inclusion of a centrifugation step to remove excess fat before testing.

#### 2.2.2. Determination of the Infant FUT2 Genotypes 

Infant FUT2 genotypes were determined using a previously published Restriction-Fragment Length Polymorphism polymerase chain reaction (PCR) method [26], on deoxyribonucleic acid (DNA) extracted from infant buffy coat using the QIA Amp^®^ DNA mini kit (QIAGEN, Hilden, Germany). Briefly, using extracted genomic DNA and previously published primers, conventional PCR was used to amplify the FUT2 gene and amplicons were confirmed via the electrophoresis of PCR products on 1.5% agarose gel [27]. Bands were visualised under ultra-violet (UV) light alongside a molecular marker. Purified DNA amplicons were then used to perform restriction fragment length polymorphism (RFLP) PCR with the AvaII enzyme. The restriction fragment length polymorphism (RFLP) PCR reaction was carried out with AvaII (Thermo Scientific^®^, Vilnius, Lithuania), and products of restriction enzyme digestion were electrophoresed and visualised under UV light. FUT2 genotypes were determined based on RFLP patterns (Appendix A).

#### 2.2.3. Measurement of Rotavirus-Specific IgA 

A validated sandwich ELISA assay was used to measure rotavirus-specific immunoglobulin A (RV-IgA) in infant plasma samples, as described previously [24]. The assay employs the use of mock-infected African green monkey kidney (MA104) cells and rotavirus WC3 strain viral lysates. Standard serum with assigned RV-IgA U/mL obtained from the Laboratory for Specialized Clinical Studies, Cincinnati Children’s Hospital Medical Centre (CCHMC), Cincinnati, Ohio, USA, was used to generate and validate an in-house plasma assay standard pooled from ROTARIX^®^-vaccinated adult donor volunteers. Absorbance was read at 492 nm using a Gen5 software (version 2.0)-enabled EPOCH^TM^ 2 microplate reader (Agilent, Santa Clara, CA, USA), and outputs were read as rotavirus-specific IgA titres in U/mL.

### 2.3. Statistical Analysis

To assess the relationship between maternal and infant HBGA profiles and Rotarix^®^ immunogenicity with a 95% confidence level, a sample size of 135 participants was required. Based on a previously reported seroconversion rate of 60.2% [10], a confidence interval of 95%, and a precision of 5% (adjusting for the finite population in the main RCT), a sample size of 135 was obtained using the Cochrane formula. We used simple random sampling to assign random numbers to our sorted IDs in the sampling frame (study participants’ IDs from the parent study) after we ‘set seed’ for replication purposes. The random numbers were then sorted in ascending order based on the assigned random number. We then picked the first 135 ordered numbers to obtain our sample size.

Participants’ socio-demographic characteristics were summarised as proportions and means (standard deviations)/medians (interquartile range) depending on the distribution of the data. A chi-squared or Fisher’s exact test was used to determine the association between categorical variables and seroconversion. A *t*-test and analysis of variance (ANOVA) were used to compare geometric mean RV-IgA titres, at each time point, between groups and among groups, respectively. To estimate the geometric mean ratio (GMR) and accompanying confidence intervals, simple linear regression was performed on log-transformed (on the natural log scale) RV-IgA titres. Seroconversion was defined as a four-fold increase or greater in serum RV-IgA titre between pre-vaccination and one month post-dose-2 ROTARIX^®^ vaccination [24]. We assessed the crude effect of children’s baseline characteristics on seroconversion using logistic regression. Statistical significance was set at a *p*-value < 0.05. All statistical analyses were performed using Stata version 17 (Stata Corp, College Station, TX, USA). 

## 3. Results

### 3.1. Participant and Sampling Flow Chart

For this study, 135/212 (64%) enrolled infants were followed up for phenotyping analysis, and 90/135 were randomly selected for FUT2 genotyping to determine their secretor genotypes. Participants who had no corresponding rotavirus-IgA data at the 3-, 9-, and 12-month time points due to study dropouts were not included in the final analysis, as shown in the flow chart (Figure 1).

### 3.2. Study Population Characteristics and Overall Seroconversion Frequency

The Median age of infants was 6 weeks (IQR 6-6), with a higher proportion of males (53.9%, *n* = 69) than females (46.1%, *n* = 59) (Table 1). A total of 71 (55.5%) participants were randomised to the intervention arm, while the rest were in the control arm of the main study. One hundred and twenty-two (95.3%) infants were exclusively breastfed and 39 (30.5%) were HIV-exposed. Among the children, 21 (16.4%) of the children were stunted, 9 (7.0%) children were wasting, and 2 (1.6%) were malnourished at enrolment. Most infants came from households with shared toilet facilities (81.3%, *n* = 104) and utilised a public tap, pipe water, or borehole (60.9%, *n* = 78), and most mothers had attained a secondary level of education (63.3%, *n* = 81). Only 29% of infants enrolled in the study seroconverted, and seroconversion was not statistically associated with any of the infants’ or mothers’ baseline characteristics (Table 1).

### 3.3. Mother and Infant HBGA Profiles

The frequency of the maternal Lewis-positive phenotype was 83 (64.8%), while that of the Lewis-null phenotype was 45 (35.2%). The frequencies of secretors and non-secretors were 22 (17.2%) and 106 (82.8%), respectively (Figure 2a). Among the infants’ ABO phenotypes, group O had a frequency of 103 (80.5%), followed by group A at 22 (17.2%). The frequency of the Lewis-positive phenotype (Le+) was 64.1% (*n* = 82), while that of the Lewis-null (Le−) phenotype was 35.9% (*n* = 46). Similarly, there was a higher frequency of secretors (Se) (81.3%, *n* = 104) compared to non-secretors (se) (18.8%, *n* = 24). In the subset of infants (*n* = 90) on which FUT2 genotypes were determined, the frequency of homozygous secretors (GG) was 73.1%, while that of heterozygous secretors (GA) was 4.9% and of non-secretors was 22% (Figure 2b). 

### 3.4. Maternal and Infant HBGA and RV-IgA Immunogenicity

We plotted the trends of the RV-IgA titres of infants from pre-vaccination to 3 months post-third dose of ROTARIX^®^ for the ABO and Lewis phenotypes. There was no observable significant difference in mean RV-IgA titres across ABO phenotypes at the baseline, post dose-2, and pre-dose 3 time-points (Figure 3a). However, a significant difference in mean titres was observed for the ABO phenotype at post-dose 3. We also observed a significant difference in the mean titres post-dose 3 for the Lewis phenotypes and the secretor phenotype (Appendix A).

Using one-way ANOVA, we tested for associations of infant and maternal HBGAs with the RV-IgA titre 1 month post-second dose of ROTARIX^®^, which was our seroconversion determination time-point. We found that infant ABO, Lewis, and secretor status were not associated with geometric mean titres (GMTs) 1 month post-second dose (*p* = 0.874, *p* = 0.332 and *p* = 0.279), respectively. Both maternal Lewis and secretor phenotypes were not associated with GMTs (*p* = 0.358 and *p* = 0.850). Similarly, no statistically significant difference was observed in the geometric mean titre ratio (GMR) for all maternal and infant HBGA profiles (*p* ≥ 0.05) (Table 2). We further used Chi-square tests to determine whether the seroconversion frequency varied across HBGA phenotypes, and we found that there was no statistically significant difference in seroconversion observed across infant ABO phenotype (*p* = 0.929), Lewis phenotype (*p* = 0.775), secretor phenotype (*p* = 0.24), and secretor genotype (*p* = 0.289), and we could not adjust for background characteristics since no variables showed significantly lower/higher crude odds of seroconversion (Table 2).

### 3.5. Maternal and Infant HBGA and RV-IgA Immunogenicity 3 Months Post-Dose-3

We performed a one-way ANOVA analysis to determine the effect of HBGAs on rotavirus-IgA geometric mean titres (GMT) in infants at 12 months of age, 3 months after the third dose of ROTARIX^®^, for those in the intervention arm (Table 3). As our seroconversion definition could not be used at the 12-month time-point, we used simple linear regression to compute the GMT ratios. We observed a significant association between ABO phenotype and GMTs (*p* = 0.02), with lower GMTs observed in group O (3.7 (95% CI: 0.35, 4.08)), compared to group AB (5.28 (95% CI: 1.86, 15)), and group A (5.02 (95% CI: 4.14, 6.07)) (Table 3). The infant Lewis-positive phenotype Le+ (Le a+b−, Le a−b+, or Le a+b+) had significantly higher GMTs (4.17 (95% CI: 3.75, 4.64) vs. 3.57 (95% CI: 3.03, 4.22)) than the Lewis-null phenotype (Le a−b−); (*p* = 0.015). We also observed significantly higher GMTs in the infant secretor phenotype compared to non-secretors (4.14 (95% CI: 3.78, 4.54) vs. 2.89 (95% CI: 2.26, 3.71)); *p* < 0.001. Infant secretor genotype, maternal Lewis phenotype, maternal secretor phenotype, and treatment arm were not significantly associated with GMTs (*p* = 0.521, *p* = 0.368, and *p* = 0.26), respectively. The ABO phenotypes showed no significant differences in GMT ratio for group AB *(p* = 0.560) and group A (*p* = 0.14), respectively. However, group AB had higher GMTs, followed by group A (5.02 (95% CI: 4.14, 6.07)), with the group with the least being group O (3.7 (96% CI: 3.35, 4.08)); *p* = 0.002. Similarly for the Lewis phenotype, the Lewis-positive phenotype Le+ (Le a+b−, Le a−b+, or Le a+b+) showed significantly higher GMTs (4.17 (95% CI: 3.75, 4.63)) compared to the null phenotype (3.57 (95% CI: 3.03, 4.22)); *p* = 0.002. Furthermore, at the phenotype level, secretors showed a statistically higher GMT ratio compared to non-secretors (*p*< 0.001), though genotype (FUT2) was not found to be statistically significant (*p* = 0.063). Maternal Lewis phenotype and maternal secretor phenotype were not significant for both GMTs (*p* = 0.521, *p* = 0.368) and the GMT ratio (*p* = 0.863, *p* = 0.751), respectively (Table 3).

A third booster dose of ROTARIX^®^ at 9 months showed no significant effect on immunogenicity between the control and intervention arms at 12 months for both GMTs and the GMT ratio (*p* = 0.26 and *p* = 0.479), respectively All other HBGA variables were not significantly associated with the GMT ratio (Table 3).

From the parent study, only 4 (5.3%) out of 76 stool samples passively collected from participants presenting with diarrhoea, and 3 were positive for rotavirus (two G3, one G4 genotypes, while 1 sample was untypable due to insufficient volume) [24].

## 4. Discussion

To the best of our knowledge, this is the first study conducted in Zambia that has attempted to assess both genotypic and phenotypic secretor effects on Rotarix^®^ immunogenicity in U5 children and that accounts for the influence of maternal profiles. Our study investigating the influence of maternal and infant histo-blood group antigens on Rotarix^®^ immunogenicity yielded three main findings: (i) There was no association between maternal and infant HBGAs on vaccine immunogenicity at 1 month post-second dose; (ii) Maternal HBGAs had no effect on vaccine immunogenicity in infants at 12 months of age; and (iii) infant HBGAs are associated with immunogenicity much later in life at 12 months of age. These findings both correlate with and contradict various publications on this subject.

Several studies have shown that HBGAs are important in host–pathogen interactions, and their potential role in infection and vaccine uptake has been hypothesised [17,22,23,28,29,30,31,32]. Our findings that HBGAs were not significantly associated with vaccine immunogenicity in early infant life (Table 2) were very similar to those from a study in neighbouring Malawi, which found no association of ABO, Lewis, and secretor status with seroconversion or vaccine shedding in early post-vaccination [13]. The same study showed high concordance of secretor genotype and phenotype proportions, though neither profile was found to influence immunogenicity post-vaccination, as in our study [13]. Contrary to these findings, a study conducted in Nicaragua showed that ABO blood groups seem to be significantly associated with rotavirus vaccine immunogenicity [14], as was shown in our study in infants at 12 months of age with significantly varied GMTs, GMRs, and GMFR (Table 3) reported across ABO and secretor phenotypes [14]. These data support the hypothesis that HBGAs impact vaccine uptake in children and consequently impact immunogenicity, similar to studies conducted elsewhere [15,33,34]. We postulate that the observed difference in immunogenicity between Sub-Saharan Africa and North American countries might be due to the inherent genetic polymorphisms that dictate the different phenotypic profile characteristic of these unique populations. Evolutionary, selective pressure acting on pathogens might also influence susceptibility through host-range specificity, which may influence vaccine immune responses, depending on the prevalence of phenotypes in a particular population.

Of note, our study showed a higher rate of seroconversion among secretors compared to non-secretors (Table 2), as previously documented [26,35,36]. This is not surprising as the literature has shown that secretors express HBGAs on their gut-mucosal epithelia, which serve as receptors for 2 attachment—in this case, a vaccine-derived live-attenuated virus—which could explain the higher immune responses seen in secretors compared to non-secretors. We also showed consistency between secretor genotype and phenotype and immunogenicity at 12 months, which further affirms our findings and strengthens our confidence in the observed outcomes. 

In addition to secretor status, the Lewis phenotype has also been shown to impact vaccine efficacy, as reported elsewhere [35,37]. Other studies have further associated secretor and Lewis phenotypes with RV-diarrhoea through other mechanisms such as the modulation of the gut microbiota [36], while other researchers hypothesise an influence on infant gut microbiota, which influences vaccine immunogenicity [38,39].

While we may not fully understand the role malnutrition plays in vaccine immunogenicity, it has been hypothesised that a lack of essential micronutrients (e.g., iron, zinc, and vitamin A) impairs IgA antibody production, specific T-cell-mediated production and gut barrier function [23,30]. We, therefore, think this is a possible explanation for the observed effect on ROTARIX^®^ immunogenicity in our study.

Though our study only focused on the association, other studies investigated the interaction between HBGAs and rotaviruses at a molecular level, which has been shown to occur in a genotype-specific fashion. Using Nuclear Magnetic Resonance techniques (NMR), one study found that A-type antigens were recognised as receptors for human rotaviruses, while the human (P8) rotavirus Wa strain did not recognise A-type HBGAs [40]. Meanwhile, another similar study showed that rotavirus genogroups (P4) (P6) and (P8) of the VP8* subunits recognised Lewis-b and/H-type-1 antigens and are therefore important factors to be considered in the production of P-type-based vaccines [34]. More studies have shown that rotaviruses have host-range specificity based on the prevalence of certain HBGA phenotypes. This varies from region to region, thereby influencing both strain diversity and host susceptibility, which is an important evolutionary attribute [17,28,32,41,42,43]. Studies conducted to assess host-genetic susceptibility via HBGA and vaccine immunogenicity strongly suggest that this relationship could partly explain why vaccine efficacy is poor in LMICs compared to high-income countries (HIC) [42,43]. 

We find the contrasting immunogenicity pictures at 1 month post-second dose (Table 2) and 12 months (Table 3) in our study to be a very interesting phenomenon worthy of more attention. We agree with the theory that there is more influence of maternal factors, such as maternal immunity and the non-immunogenic components of breast milk interfering with infants’ immune responses, as shown by previous studies conducted in the country [9,10]. Importantly, we observed from our trend plots that at 9 months, RV-IgA titres were significantly higher than at all the earlier time points, even before the intervention arm received the third booster dose of Rotarix^®^. Therefore, the significant increases seen in titres at 12 months were most likely due to natural exposure to wild-type rotavirus, given that there was no statistically significant difference observed in titres between the intervention and control arms of the study. It is therefore plausible that the AB and A group receptors may have been interacting with wild-type rotavirus through the VP8 subunit in a “type-specific manner”, as documented elsewhere [41,42,43].

Though no association was found between maternal HBGAs and infant RV-IgA titres in our results (Table 2 and Table 3), we find it plausible that secreted HBGAs in mothers’ breast milk could serve as decoy receptors and thereby limit the available fraction of vaccine material to be actively taken up by infants, thereby resulting in a lack of seroconverters and insignificant GMR observed 1 month post-second dose. This phenomenon also feeds into another hypothesised theory of a developmental delay in the biosynthesis of HBGAs, stating that in early infant life, HBGAs are not fully expressed in gut-mucosal epithelial tissue and, therefore, there are fewer attachment sites for the vaccine-derived virus, leading to sub-optimal uptake. However, as the child grows and expresses more HBGAs in their body tissue and is gradually weaned off breast milk, RV-IgA titres seem to show an increase, as seen at 12 months in our study cohort. It is, however, unclear whether this increase in titres is due to a delayed effect of vaccination or attributable to wild-type infections. However, HBGAs could likely play a part in titre kinetics.

Though our study showed no effect of maternal HBGAs on vaccine immunogenicity, a study elsewhere reported a higher seroconversion frequency in children born to non-secretor mothers compared to secretor-positive mothers [21]. Interestingly, this study, like ours, found that infant Lewis and secretor phenotypes were not associated with seroconversion at 18 weeks [21]. The working hypothesis is that children born to non-secretor mothers have a reduced risk of interference compared to those born to secretor-positive mothers who shed decoy receptors in their breast milk.

We also note that the Lewis-null phenotype (non-secretor) had very low immunogenicity measures even in infants at 12 months of age (Table 3). This might be due to the host-range specificity that has been shown regarding rotavirus in the literature. It is well documented that most (P4), (P6), and (P8) human rotaviruses recognise H-type 1 and Lewis-B antigens [11,17,21,41], and it is therefore plausible that mothers with the Lewis-null (non-secretors) phenotype had no receptors for the vaccine-derived G1(P8) strain, leading to the low vaccine response observed.

The strengths of our study were that our study population was drawn using a randomised controlled trial, and hence, has a reduced risk of bias, as well as being accorded the statistical power to control for confounding variables. Employing both phenotypic and genotypic methods for infant secretor genotyping also strengthened our interpretation of the results for our outcome variable. Our study, however, also had several limitations. Firstly, our sample size was small, hence the wide confidence intervals, which could also have been influenced by wide variation within our study population. A larger cohort study and a longer follow-up period would be ideal to measure the effect size of our outcome variable accurately. We did not assess the confounding effect of maternal antibodies on immunogenicity in this study. The use of more phenotypic than molecular techniques, which are more robust, also reduced the sensitivity of our assays. It would also be important to conduct this study in children presenting with diarrhoea where aetiology and vaccine shedding can be assessed in addition to serological work.

## 5. Conclusions

In summary, this study found that in general, HBGAs were not associated with ROTARIX^®^ immunogenicity. We recommend that future research be focused on understanding the full extent of this influence, which will inform the design of more efficacious vaccines that will bypass this gut–mucosal barrier and have improved efficacy in LMICs. Such studies will help inform policy on strategies aimed at improving vaccination outcomes in U5 children and consequently improve their health status.

It is also critical to set up surveillance systems that will monitor the molecular epidemiology of wild-type rotaviruses since the introduction of ROTARIX^®^ to monitor the evolutionary patterns occurring in nature. This will enable the idealization of appropriate interventions and enable us to move toward a more proactive approach targeted at eliminating rotavirus soon.

## Figures and Tables

**Figure 1 vaccines-11-01303-f001:**
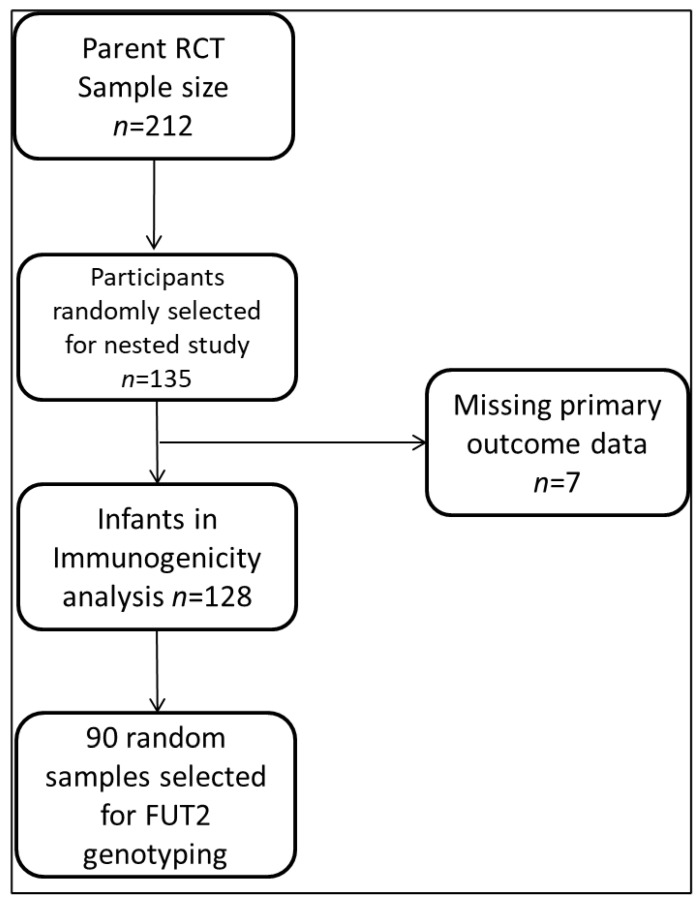
Analysis flow chart.

**Figure 2 vaccines-11-01303-f002:**
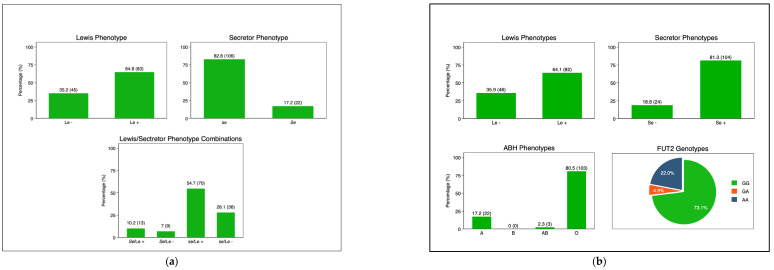
(**a**). Maternal HBGA profiles: secretor status: secretor (Se) and non-secretor (se); Lewis phenotype: Lewis-positive (Le+) and Lewis-null (Le−). (**b**). Infant HBGA profiles; ABH: A, B, AB, and O; secretor status: secretor (Se) and non-secretor (se); Lewis phenotype: Lewis-positive (Le+) and Lewis-null (Le−). Infant FUT2 Genotypes: GG: homozygous secretor; GA: heterozygous secretor; and AA: non-secretors. Abbreviations: Le-Lewis; Se-secretor; se-non-secretor; FUT2-fuscosyltransferase-2.

**Figure 3 vaccines-11-01303-f003:**
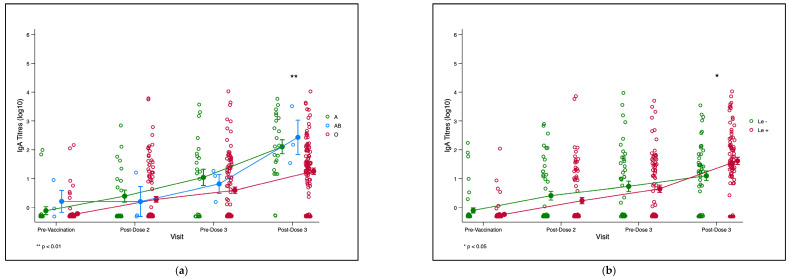
(**a**). Trend plot for infant RV-IgA titre kinetics for ABO phenotype. The green, blue and red lines represent trends of RV-IgA titres over time for blood groups A, AB, and O, respectively.** *p* = 0.002. (**b**). Trend plot for infant RV-IgA titre kinetics for Lewis phenotype. The green and red lines represent the trends of RV-IgA titres over time for the Lewis-null (Le−) and Lewis-positive (Le+) phenotypes, respectively.* *p* = 0.015. (**a**,**b**). Trends in infant rotavirus-specific immunoglobulin A (RV-IgA) titres pre- and post-rotavirus vaccination compared infant and maternal HBGA profiles. Each circle represents an infant’s log10 RV-IgA titre. Yellow circles with lines represent means and standard errors of log-transformed RV-IgA titres. Abbreviations: Le-Lewis.

**Table 1 vaccines-11-01303-t001:** Mother and infant baseline characteristics and seroconversion status 1 month after ROTARIX^®^ dose 2.

		Seroconverted		
	Mother–Infant Pairs (*N* = 128)	No (*n* = 91, 71.1%)	Yes (*n* = 37, 28.9%)	*p*-Value
	*n* (% of total)	*n* (%)	*n* (%)	
Infants’ Characteristics				
Age (Weeks)				
Median (IQR)	6 (6–6)	6 (6–6)	6 (6–6)	0.442
Mean (SD)	6 (0.6)	6 (0.6)	5.9 (0.7)	
Sex				
Male	69 (53.9)	51 (73.9)	18 (26.0)	0.447
Female	59 (46.1)	40 (67.7)	19 (32.2)	
Feeding				
Exclusively breastfeeding	122 (95.3)	86 (70.4)	36 (29.5)	0.672
Mixed feeding	6 (4.7)	5 (83.3)	1 (16.6)	
Birthweight (kg)				
<2.5	5 (3.9)	3 (60.0)	2 (40.0)	0.626
≥2.5	123 (96.1)	88 (71.5)	35 (28.4)	
HIV Exposure				
Not exposed	89 (69.5)	62 (69.6)	27 (30.3)	0.590
Exposed	39 (30.5)	28 (73.6)	10 (26.3)	
Nutritional Status				
Stunted				
No (HAZ ≥ −2)	107 (83.6)	78 (72.8)	29 (27.1)	0.310
Yes (HAZ < −2)	21 (16.4)	13 (61.9)	8 (38.0)	
Wasting				
No (WAZ ≥ −2)	119 (93.0)	86 (72.2)	33 (27.7)	0.281
Yes (WAZ < −2)	9 (7.0)	5 (55.5)	4 (44.4)	
Mothers’ Characteristics				
Age (years)				
<20	20 (15.6)	15 (75.0)	5 (25.0)	0.080
20–24	45 (35.2)	37 (82.2)	8 (17.7)	
25–29	34 (26.6)	19 (55.8)	15 (44.1)	
≥30	29 (22.7)	20 (68.9)	9 (31.0)	
Highest Education Level				
None	6 (4.7)	4 (66.7)	2 (33.3)	0.470 *
Primary	40 (31.3)	25 (62.5)	15 (37.5)	
Secondary	81 (63.3)	61 (75.3)	20 (24.6)	
Tertiary	1 (0.8)	1 (100.0)	0 (0.0)	
Water Source				
Piped into house/yard	45 (35.2)	33 (75.0)	12 (25.0)	0.882
Protected well	5 (3.9)	4 (80.0)	1 (20.0)	
Public borehole/tap and pipe	78 (60.9)	54 (80.0)	24 (20.0)	
Shared Toilet Facility				
No	24 (18.8)	17 (70.8)	7 (29.1)	0.975
Yes	104 (81.3)	74 (71.1)	30 (28.8)	
Type of Toilet Facility				
Flushing toilet	26 (20.3)	17 (65.4)	9 (34.6)	0.476
Pit latrine	102 (79.7)	74 (72.6)	28 (27.5)	

Abbreviations: IQR—interquartile range; SD—standard deviation; Kg—kilogram; HAZ—height-for-age Z-score; HIV—human immunodeficiency virus; MR—measles–rubella vaccine; RV-IgA—rotavirus specific immunoglobulin A; WAZ—weight-for-age Z-score; WLZ—weight-for-length Z-score. * Fisher’s exact test.

**Table 2 vaccines-11-01303-t002:** Maternal and infant HBGA profiles and anti-rotavirus IgA titres 1 month post-ROTARIX^®^ dose 2.

Characteristics	Number of Mother–Infant Pairs (% of Total)	GMTs (95% CI)	ANOVA *p*-Value	Seroconversion (*n* = 37, 28.9%)	Chi-Square *p*-Value	Crude Odds Ratio (95% CI)	*p*-Value
				*n* (%)			
Infant							
Infant HBGA Phenotype							
A	22 (17.2)	2.5 (0.9, 6.8)	0.874	7 (31.8)	0.929	ref	
AB	3 (2.3)	1.6 (0, 270.6)		1 (33.3)		1.1 (0.1, 13.9)	0.958
O	103 (80.5)	1.9 (1.2, 3)		29 (28.2)		0.8 (0.3, 2.3)	0.731
Infant Lewis Phenotype							
Le− (Le a−b−)	46 (35.9)	2.6 (1.3, 5.2)	0.332	14 (30.4)	0.775	ref	
Le+ (Le a+b−, Le a−b+, or Le a+b+)	82 (64.1)	1.7 (1.1, 2.8)		23 (28.2)		0.9 (0.4, 2)	0.775
Secretor Phenotype							
Non-secretor (se)	24 (18.8)	1.3 (0.6, 2.8)	0.279	5 (20.8)	0.24	ref	
Secretor Phenotype (Se)	104 (81.3)	2.2 (1.4, 3.5)		32 (30.8)		1.7 (0.6, 4.9)	0.337
Infant FUT2 Genotype *							
Homozygous secretor (GG)	60 (46.9)	1.4 (0.8, 2.5)	0.093	15 (25.0)	0.289	ref	
Heterozygous secretor (GA)	4 (3.1)	5.6 (0, 1426.5)		2 (50.0)		3 (0.4, 23.2)	0.292
Non-secretor (AA)	18 (14.1)	4.9 (1.5, 16.3)		7 (38.9)		1.9 (0.6, 5.8)	0.255
Missing	46 (35.9)	2 (1, 3.8)		13 (28.3)		-	-
Mother							
Lewis Phenotype							
Le− (Le a−b−)	45 (35.2)	1.6 (0.9, 2.8)	0.358	13 (28.9)	0.997	ref	
Le+ (Le a+b−, Le a−b+, or Le a+b+)	83 (64.8)	2.3 (1.4, 3.9)		24 (28.9)		1.0 (0.4, 2.2)	0.997
Secretor Phenotype							
Non-secretor (se)	106 (82.8)	2 (1.3, 3.1)	0.85	32 (30.2)	0.336	ref	
Secretor Phenotype (Se)	22 (17.2)	1.8 (0.7, 5.2)		5 (22.7)		0.7 (0.2, 2.0)	0.484

* Individuals with the genotype GG or GA at position 428 of the FUT2 gene are called homozygous and heterozygous secretors (Se), respectively. The G428A mutation in the FUT2 gene gives rise to an early stop codon, giving a truncated non-functional protein. Homozygous carriers of a nonsense mutation (AA) in this gene are called non-secretors (se). Abbreviations: GMT—Geometric mean titres; GMR—Geometric mean ratio; GMFR—rise; Le—Lewis; Se—secretor.

**Table 3 vaccines-11-01303-t003:** Maternal and infant HBGA profiles and anti-rotavirus IgA titres at 12 months.

Characteristics	V12 GMTs	ANOVA, *p*-Value	GMT Ratio (95% CI)	*p*-Value
	GMT (95% CI)			
Infant				
Infant ABO Phenotype				
A	5.02 (4.14, 6.07)	0.002	ref	
AB	5.28 (1.86, 15)		0.59 (0.10, 3.47)	0.560
O	3.7 (3.35, 4.08)		0.36 (0.09, 1.41)	0.140
Infant Lewis Phenotype				
Le− (Le a−b−)	3.57 (3.03, 4.22)	0.015	ref	
Le+ (Le a+b−, Le a−b+, or Le a+b+)	4.17 (3.75, 4.63)		0.83 (0.31, 2.23)	0.705
Secretor Phenotype				
Non-secretor (se)	2.89 (2.26, 3.71)	<0.001	ref	
Secretor phenotype (Se)	4.14 (3.78, 4.54)		1.94 (0.59, 6.4)	0.276
Infant FUT2 Genotype				
Secretor (GG)/(GA)	3.95 (3.45, 4.52)	0.063	ref	
Non-secretor (AA)	3.24 (2.44, 4.31)		1.66 (0.96, 2.83)	0.543
Mother				
Lewis phenotype				
Le− (Le a−b−)	4.02 (3.52, 4.58)	0.521	ref	
Le+ (Le a+b−, Le a−b+, or Le a+b+)	3.95 (3.51, 4.44)		1.09 (0.41, 2.88)	0.863
Secretor Phenotype				
Non-secretor (se)	4.08 (3.72, 4.48)	0.368	ref	
Secretor Phenotype (Se)	3.45 (2.64, 4.51)		0.83 (0.25, 2.70)	0.751
Treatment Arm				
Control (MR)	4.08 (3.56, 4.67)	0.260	ref	
Intervention (ROTARIX^®^ + MR)	3.88 (3.44, 4.37)		1.39 (0.55, 3.49)	0.479

Abbreviations: GMT—geometric mean titres, Le—Lewis; Se—secretor; MR—measles–rubella.

## Data Availability

The data presented in this study are available on request from the corresponding author. The data are not publicly available due to policy restrictions on institutional data publication.

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
