# Peer review of "Maternal and Infant Histo-Blood Group Antigen (HBGA) Profiles and Their Influence on Oral Rotavirus Vaccine (RotarixTM) Immunogenicity among Infants in Zambia"

_vaccines, 2023, doi:10.3390/vaccines11081303_

Round 1

Reviewer 1 Report

This is a very important and timely study. It checked if any association exists between HBGAs and the capability of ROTARIX vaccine to induce immunological responses if and when administers to U5 children.

The authors futher suggested futher studies among which is a surveillance study to monitor molecular epidemiology of wild-type virus with possible differentiation from vaccine type.

Overall, the study is excellent. However, authors should highlight the possible reasons, if any, why their findings contrasted with some previous works done in Africa in the area.

Very minor corrections are needed throughout the manuscript. Overall, the language and reporting formats are good

Author Response

Thank you for that important suggestion. Due to limited research on this subject in Africa, we compared our findings with three(3) studies from Ghana, Burkina Faso and Malawi to be specific. In paragraph 2 of section 4 (Lines 275-282), we highlight the similarity of our findings to the cited study in neighbouring Malawi while our third conclusion that HBGAs are associated with vaccine efficacy much later in life are consistent with the reports in Ghana and Burkina Faso (Lines 275-282) while the study in Tunisia confirms our hypothesis that the P8 vaccine strain interacts with HBGAs and may impact vaccine efficacy. We also highlight our thoughts in paragraphs 6, 7 and 8 of section 4 regarding the possible explanation for the contrasting results and hope this addresses the comments adequately (Lines 327-252).

Reviewer 2 Report

The study investigated whether maternal and infant intrinsic susceptibility to rotavirus infection via histo-blood group antigen (HBGA) profiles influenced rotavirus (ROTARIX®) vaccine induced responses in Zambia. They aimed to profile the maternal and infant HBGA phenotypes and genotypes, and determine their influence on ROTARIX immunogenicity in a mother infant pair cohort in Zambia. The study was well executed and address the research question especial for developing countries and Rota vaccine immunogenicity.

This is an important topic especially in the enteric field especially with Rota vaccine immunogenicity diminishing in low income countries. The study is original, relevant especially is sub Saharan Africa more countries are implementing Rotavirus vaccine.

The study adds in the field of vaccinology, Rota vaccine immunogenicity diminishing in low income countries, and vaccine responses in different geographic areas and genetic make -up. While this study only focus on Rotavirus, other research focusing on viruses like Norovirus may benefit especially with histo-blood group antigen influence and maternal antibodies.

It's well written and the recommendation are within the findings. As such the future research be focused on understanding the full extent of this influence, which will inform the design of more efficacious vaccines that will by-pass this gut-mucosal barrier and improve efficacy in LMICs as per study

Author Response

We are grateful for taking the time to review our work and appreciate the comments given.

Reviewer 3 Report

Maternal and infant histo-blood group antigens (HBGA) profiles and it’s influence on oral rotavirus vaccine (ROTARIX®) immunogenicity among infants in Zambia..

The authors have investigated whether maternal and infant intrinsic susceptibility to rotavirus infection via histo-blood group antigen (HBGA) profiles influenced rotavirus (ROTARIX®) vaccine-induced responses in Zambia.

The article is well-researched and presented in a very understandable and informative way. This article is highly impactful and can be accepted in its present form.  

Minor editing of English language required

Author Response

Thank you for taking time to review our work and for the suggestions given.
We have since run some revisions of the manuscript text in our write-up and hopefully addressed your suggestion adequately.

Reviewer 4 Report

The manuscript reports a prospective cohort study aimed at investigating the influence of maternal and infant intrinsic susceptibility to rotavirus infection via histo-blood group antigen (HBGA) on rotavirus (ROTARIX®) vaccine-induced responses in Zambia. Although the topic is relevant and the strengths/limations of the study are welll recognized by the authors, some points requires revision and are listed below in the order of appearance in the text:

- Lines 95-97: Detail conditions for ELISA reactions, including precise incubation times and antibody sources.

- Line 107: Provide the exact publication where the used primers and PCR cycling conditions were originally described.

- Lines 163-164: Correct the statement "39 (30.5%) [infants] were HIV-unexposed" since this is the number/percentage of HIV-exposed infants according to Table 1.

- Lines 166-167: Correct the statement "most infants [...] utilised a public tap, pipe water ou borehole (3.9%, n=78)" since this is the percentage of infants using protected well as water source according to Table 1.

- Lines 185-188: Revise the description of FUT2 genotype data for a proper correspondence with the data shown in the related graph of Figure 2b.

- Lines 197-198: Provide the data related to the mean RV-IgA titres at post-dose 3 for the secretor phenotype.

- Line 199: Use asterisks to indicate statistically significant differences in Figures 3a and 3b.

- Line 207: Correct word repetitions in the sentence "we could not we could not adjust for background characteristics".

- Lines 223-225: Correct data repetition for group O ("3.7 [95% CI, (3.35, 4.08)]") and indication of Table 4 instead of Table 3.

- Lines 231-238: Correct information repetition in the sentences "The ABO phenotypes showed no significant differences in GMT ratio for group AB (p=0.560) and group A (p=0.14) respectively.followed by group A 5.02[95% CI, (4.14, 6.07)] and the least being group O 3.7[96% CI, (3.35, 4.08)]; p=0.002. Similarly among the Lewis phenotype, Lewis positive phenotype Le+ (Le a+b-, Le a-b+, or Le a+b+) showed significantly higher GMTs 4.17 [95% CI, (3.75, 4.63)} compared to the null phenotype 3.57 [95% CI, (3.03, 4.22)]; p=0.002.. Furthermore, at phenotype level, secretors showed a statistically higher GMT ratio compared to non-secretors (p< 0.001)".

- Lines 239-240: Correct the p-values for the difference between adjusted and undjusted GMT ratio of maternal Lewis phenotype and maternal secretor phenotype.

- Line 323: Insert "found" after "was" in the sentence "Though no association was between maternal HBGAs".

Minor editing of English language required as indicated above.

Author Response

Dear Reviewer 4, please see the attachment.

Reviewer 5 Report

In the paper entitled " Maternal and infant histo-blood group antigens (HBGA) profiles and it’s influence on oral rotavirus vaccine ROTARIX® immunogenicity among infants in Zambia. ", Chauwa and co-authors study the mother and infant HBGA profiles and how it influences ROTARIX® immunogenicity in U5 infants. They utilized ELISA, and PCR for experiments in the support of their claim. Based on the observations- they found that there is no association between maternal and infant HBGAs on vaccine immunogenicity (which figure or table), maternal HBGAs have no effects on vaccine immunogenicity, however, HBGAs have some considerable role in immunogenicity after 12 months of infant age.

Authors only considered the safety and immunogenicity after 3rd booster dose of ROTARIX® delivered at 9 months of age (line 74), however I couldn’t understand the recommendation of CDC for ROTARIX® only for two doses at 2 months and 4 months. This point should be included in the manuscript to avoid doubt. Another missing data that needs to correlate the efficacy of vaccination and the incidence of diarrhoea (none of the infants show diarrhea after vaccination?) due to rotavirus. Further, authors selected 135 participants and determine the WC3 specific IgA antibody at various time points and correlated with HBGA profiles, however correlation plot is missing (Fig3 a and b).

Another point which is important was to compare or to present data- group of mothers based on antibody and HBGAs phenotype and their infant’s response for ROTARIX®. Since the rotavirus specific IgA level is missing for all the pre and post vaccination time, with the HBGAs profiles as it will also varies with time. The authors already included their drawback of this study but children with diarrhoea and vaccine shedding is important to include in this study.

Finally, this finding suggests that HBGA is not significantly associated with vaccine immunogenicity specially in early infants’ life but has role after 12 months (what could be the possible explanation). Correct in conclusion section (line 364-365) that in general HBGAs are not associated with ROTARIX® immunogenicity.

Major points:

1.     Figures are confusing- plot mean value and p value in the graph itself.

2.     In discussion justify your findings by citing figures.

3.     There is no data for Fut3 though it is mentioned in the method section.

4.     Include PCR gels in the supplementary figures.

5.     The manuscript can be presented in a better way for the readerships of Vaccines.

English language is fine

Round 2

Reviewer 5 Report

I am not able to edit and submit this report on mdpi account as suggested by you as well which is likely.

Chauwa et al. address my concerns with limitations (The authors admitted that some of the experiments can't be done due to resource limitations but I think those are crucial for this paper as well clarity to the readers), provided this study is important to the field, I would recommend accepting this article in the present form.